# Role of Environment Variables in Spatial Distribution of Soil C, N, P Ecological Stoichiometry in the Typical Black Soil Region of Northeast China

**Qianqian Chen [1], Zhou Shi [1], Songchao Chen [2], Yuxuan Gou [3] and Zhiqing Zhuo [1,\*]**

1    Institute of Agricultural Remote Sensing and Information Technology Application, Zhejiang University, Hangzhou 310058, China; qqchen@zju.edu.cn (Q.C.); shizhou@zju.edu.cn (Z.S.)
2    ZJU-Hangzhou Global Scientific and Technological Innovation Center, Hangzhou 311200, China; chensongchao@zju.edu.cn
3    Key Laboratory of Agricultural Land Quality, Ministry of Natural Resources, College of Land Science and Technology, China Agricultural University, Beijing 100193, China; yuxuangou@cau.edu.cn
\*    Correspondence: zhiqingzhuo@zju.edu.cn

**Abstract:** The effects of environmental factors on topsoil nutrient distribution have been extensively discussed, but it remains unclear how they affect spatial characteristics of soil carbon (C), nitrogen (N), and phosphorus (P) stoichiometry at different depths. We collected 184 soil samples in the typical black soil region of northeast China. Ordinary kriging was performed to describe the spatial distribution of soil C, N, and P eco-stoichiometry. Redundancy analysis was used to explore relationships between C:N:P ratios and physicochemical characteristics. The soil classification was studied by hierarchical cluster analysis. The mean C, N, and P contents ranged from 15.67 to 20.08 $g \cdot kg^{-1}$, 1.15 to 1.51 $g \cdot kg^{-1}$, and 0.80 to 0.90 $g \cdot kg^{-1}$ within measured depths. C, N, and P concentrations and stoichiometry increased from southwest to northeast, and the Songhua River was identified as an important transition zone. At 0–20 cm, soil water content explained most of the C, N, and P content levels and ratios in cluster 1, while latitude had the highest explanatory ability in cluster 2. For 20–40 cm, soil bulk density was the main influencing factor in both clusters. Our findings contribute to an improved knowledge of the balance and ecological interactions of C, N, and P in northeast China for its sustainability.

**Keywords:** soil stoichiometry; Mollisols; black soil region; hierarchical cluster analysis; spatial variation

## 1. Introduction

As a widely applicable discipline, ecological stoichiometry deals with the dynamics and balance of multiple chemical elements, such as carbon(C), nitrogen(N), and phosphorus(P), and their ecological interactions in ecosystems [1,2]. Soil is the carrier of numerous ecological processes, which plays a key role in plant growth, and directly affects the composition, stability, and succession of plant communities. Being the most essential elements for soil fertility and plant growth, soil carbon (C), nitrogen (N), and phosphorus (P) regulate many ecological processes in terrestrial ecosystems [3,4]. Because the nutrient elements in the soil are commonly interrelated, it is difficult to understand the overall change characteristics of soil nutrients in a given ecosystem merely by analysing content changes in any single element [5]. In this sense, examining the relative proportions and dynamic balances of soil C, N, and P represents a fundamental and effective approach for revealing soil quality evolution in ecosystems.

Recently, numerous studies have discussed the eco-stoichiometry of soil C, N, and P in terrestrial ecosystems (e.g., wetland, grassland, forest, and desert oasis). The results indicated that soil C:N can predict the nitrate leaching degree of forest soil, while the soil N:P ratio is able to reflect the degree of vegetation disturbance [6–8]. C and N levels

play a decisive role in the changing process of regional soil C, N, and P eco-stoichiometric characteristics [9,10]. Moreover, the eco-stoichiometric characteristics of soil C, N, and P also significantly impact crop growth as well as the structure and function of agricultural ecosystems [11].

Compared with other terrestrial ecosystems, the spatial heterogeneity of soil C, N, and P eco-stoichiometry in farmland ecosystems is more pronounced, mainly because they are affected by both natural processes and human activities [12]. Several studies have reported the effects of soil properties, climatic conditions, and agricultural management strategies (e.g., cultivation systems, fertilization regimes) on the eco-stoichiometry of soil C, N, and P, mainly investigated at a small scale (e.g., fields, oases, and small agricultural watersheds) [12–15]. However, in the typical black soil region of China, most studies have focused on the topsoil, while little is known about the vertical patterns in soil C, N, and P and their stoichiometric ratios along with the soil profile [16,17]. Only a few studies dealt with the quantitative analysis of the relationship between eco-stoichiometry of soil C, N, and P and the driving factors at a regional scale, especially with the effects of natural and human factors being well recognized [8,9,18].

The black soil (Mollisol) region in Northeast China, highly suitable for crop cultivation, is the major grain-producing area in the country. However, after decades of intensive cultivation, soil in this region has suffered significant degradation physically and chemically [19,20]. Generally, conservation management practices for black soil focus mainly on nutrient elements such as soil C, N, and P, but largely neglect their stoichiometric ratios, especially at different soil depths. In this study, we investigated the soil C, N, and P concentrations and their ratios at the depths of 0–20 and 20–40 cm in a typical black soil region. The main objectives were as follows: (1) to reveal the geographical patterns of soil C, N, P contents and stoichiometric ratios both spatially and vertically; (2) to explore whether the eco-stoichiometry of soil C, N, and P shows spatial clustering; (3) to reveal the relationship and its variation between the eco-stoichiometry of soil C, N, and P and environmental variables at different soil depths.

## 2. Materials and Methods

### 2.1. Study Area

A typical black soil region refers to the main distribution areas of black soil, according to the Genetic Soil Classification of China (GSCC) [21]. The study region ($38°42'$–$53°36'$ N, $115°24'$–$135°12'$ E), located in the dryland farming region, covers a total area of $9.4 \times 10^4$ km$^2$ (Figure 1). The climate is temperate continental monsoon, with a mean annual temperature varying from $-5.7$ to $4.1$ °C. Mean annual precipitation ranges between 500 and 650 mm, with more than 70% occurring between June and August. The geomorphologic landscapes in the study area are alluvial plains and terraces, mostly with an undulating to rolling topography with gentle slopes in the range of $1°$ to $5°$. Although the original vegetation is dominated by weed communities, most of the area has been reclaimed as dryland for farming for more than 100 years because of the suitable soil and climatic conditions. Grain and cash crops, such as maize (Zea mays), soybeans (Glycine max), and sugar beet (*Beta vulgaris* L.) are typically cultivated, with one crop per year. The area is a part of the Golden Corn Belt and has become an important base for the grain commodity in China.

### 2.2. Data Sources

2.2.1. Field Investigation and Soil Sampling

We applied a 15-km-grid sampling strategy which was established using the fishnet tool of ArcGIS 10.3 (ESRI, Inc., Redlands, CA, USA). The sampling method was adopted according to the size of the area and the degree of concentration. Considering the planting system, distribution area, and concentration degree, stratified sampling for soil suborders was used to select 46 geo-referenced locations (Table 1) with coordinates recorded using a hand-held GPS (Garmin Ltd., Olathe, KS, USA) in 2017. At each location, three composite soil cores (5.1 cm in diameter) were collected and pooled for each soil depth (i.e., 0–10 cm,

10–20 cm, 20–30 cm, 30–40 cm) to determine soil physicochemical properties, resulting in a total of 184 samples. No tillage or crop planting was performed during the sampling period in late April. Three undisturbed soil cores (100 cm$^3$) were collected at each soil depth to measure the mean values of soil water content (SWC) and bulk density (BD). The SWC was determined gravimetrically after oven-drying at 105 °C for 24 h until constant weight. All composite samples were air-dried at room temperature (~25 °C) and subsequently passed through a 2-mm sieve to remove large roots and stones; prior to analysis, the samples were stored in closed zip-lock plastic bags. Soil BD was measured for different depths using the standard core method with a weight-to-volume relationship. In addition, three replicate measurements of soil penetration resistance were conducted at each sampling site for depths ranging from 0 to 45 cm at a 2.5-cm interval, with an SC-900 hand-held penetrometer (Spectrum Technologies, Inc., Aurora, IL, USA) used for penetration tests aiming to identify the plough layer thickness [22]. A laser particle size analyser (OMEC INSTRUMENTS CO., LTD., LS-909E) was used to analyse the distribution of soil particle sizes. Soil pH was determined using a pH electrode at a soil:water ratio of 1:2.5. Soil organic carbon (SOC) was measured by the dichromate heating-oxidation [23], while total nitrogen (TN) and total phosphorus (TP) concentrations were measured via SFA Segmented Continuous Flow Analysis (LICA United Technology Limited, Beijing, China). The soil type maps for the study areas were inferred from the Chinese National Soil Map with a scale of 1:4,000,000, according to the latest available data produced by the Institute of Soil Science, Chinese Academy of Science (ISSCASS).

**Table 1.** General information of sampling points and cropping system in the typical black soil region of Northeast China.

| Samples | Longitude | Latitude | Crop | Samples | Longitude | Latitude | Crop |
|---|---|---|---|---|---|---|---|
| 1 | 126.423 | 45.788 | Maize | 24 | 126.567 | 46.508 | Maize |
| 2 | 127.821 | 46.235 | Maize | 25 | 126.932 | 47.068 | Maize |
| 3 | 124.798 | 48.770 | Maize/Soybean | 26 | 127.356 | 47.663 | Maize/Soybean |
| 4 | 126.207 | 48.363 | Maize | 27 | 127.012 | 47.673 | Maize/Soybean |
| 5 | 127.630 | 45.879 | Maize | 28 | 126.837 | 47.511 | Maize/Soybean |
| 6 | 124.697 | 48.177 | Maize | 29 | 126.704 | 45.467 | Maize |
| 7 | 126.539 | 46.059 | Maize | 30 | 126.555 | 46.586 | Maize |
| 8 | 127.731 | 46.975 | Maize | 31 | 125.528 | 44.147 | Maize |
| 9 | 126.416 | 48.964 | Maize/Soybean | 32 | 125.752 | 44.252 | Maize |
| 10 | 127.205 | 47.222 | Maize/Soybean | 33 | 124.863 | 43.841 | Maize |
| 11 | 126.275 | 45.284 | Maize | 34 | 125.682 | 44.677 | Maize |
| 12 | 126.157 | 47.178 | Maize | 35 | 124.895 | 43.967 | Maize |
| 13 | 126.008 | 48.964 | Maize/Soybean | 36 | 125.497 | 44.700 | Maize |
| 14 | 126.926 | 46.093 | Maize | 37 | 126.188 | 45.011 | Maize |
| 15 | 127.550 | 45.649 | Maize | 38 | 124.489 | 43.203 | Maize |
| 16 | 127.868 | 46.367 | Maize | 39 | 124.526 | 44.024 | Maize |
| 17 | 127.046 | 45.846 | Maize | 40 | 125.976 | 45.040 | Maize |
| 18 | 126.472 | 45.357 | Maize | 41 | 126.161 | 44.325 | Maize |
| 19 | 125.283 | 48.091 | Maize/Soybean | 42 | 125.561 | 43.903 | Maize |
| 20 | 126.130 | 48.030 | Maize | 43 | 126.315 | 44.851 | Maize |
| 21 | 125.802 | 47.647 | Maize/Soybean | 44 | 124.341 | 43.364 | Maize |
| 22 | 125.715 | 47.386 | Maize | 45 | 125.186 | 43.654 | Maize |
| 23 | 126.835 | 48.393 | Maize | 46 | 123.946 | 43.288 | Maize |

### 2.2.2. Environmental Data Collection

A list of environmental variables was also collected, including mean annual temperature (MAT), mean annual precipitation (MAP), digital elevation model (DEM), multiple cropping index (MCI), and plough layer thickness (PLT). The MAT and MAP data were interpolated from the annual observations in 2017 provided by the China Meteorological Data Service Center (http://data.cma.cn/en, accessed on 21 October 2020). The DEM data was derived from the Shuttle Radar Topography Mission (SRTM) dataset at the resolution

of 90 m [24]. The MCI data was obtained from the China statistical yearbooks at the county level. The PLT data was calculated from the soil penetration resistance, measured at a depth of 40 cm. All the raster data was resampled to 1 km resolution, and the values were extracted to the sampling sites.

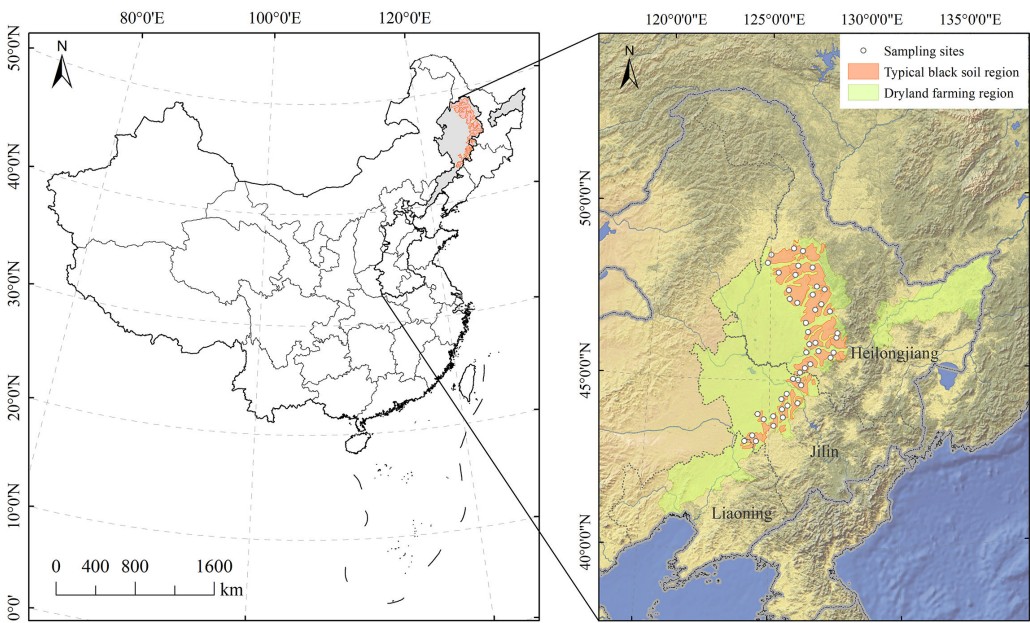

**Figure 1.** Distribution of sampling points in the typical black soil region of northeast China.

### 2.3. Calculations and Statistical Analysis

The one-way analysis of variance (one-way ANOVA) and least significant difference (LSD) test were used to estimate whether the soil C, N, and P concentrations and the elemental stoichiometry differed significantly among the soil depths ($p < 0.05$). Correlation analysis and redundancy analysis (RDA) were used to explore the relationships between soil C:N:P stoichiometry and environmental variables at different soil depths. All analyses were performed using the software package IBM SPSS Statistics (version 22, IBMSPSS, Somers, NY, USA). The assessment of hierarchical cluster compactness by a list of cluster validity indices (e.g., the CH, Duda, and McClain) and the heat maps of the correlation coefficients were conducted using the 'NbCluster' and 'corrplot' packages in R (version 4.0.5, https://www.r-project.org/, accessed on 5 June 2021) [25,26]. Spatial structure information and the distribution maps of the eco-stoichiometry of soil C, N, and P were obtained by ordinary kriging using ArcGIS 10.3.

## 3. Results

### 3.1. Spatial Distribution of Soil C, N, and P Concentrations and Eco-Stoichiometry

3.1.1. ANOVA and LSD Test Results

Considering that the 0–20 cm soil layer was more significantly affected by long-term farming and shallow ploughing than the deeper layer, the contents and stoichiometric ratios of SOC, TN, and TP were calculated for the 0–20 cm and 20–40 cm soil layers. The ANOVA and LSD test results of soil C, N, and P and other soil properties at different soil depths are listed in Table 2.

Three indicators of soil C, N, and P concentrations and stoichiometry, SOC, TN, and N:P, showed significant differences between layers. The mean contents of SOC decreased with the soil depth, with 20.08 g·kg$^{-1}$ and 15.67 g·kg$^{-1}$ at 0–20 cm and 20–40 cm, respectively. The TN and TP concentrations showed similar descending trends as the depth increased. All of the C:N, N:P, and C:P ratios were higher in the surface layer, from 13.89, 23.15, and 1.76 at 0–20 cm to 13.61, 20.49, and 1.50 at 20–40 cm, respectively. However, the distributions of mean pH, clay, BD, and SWC were the opposite. Most of the contents

and stoichiometry of soil C, N, and P showed a moderate variation [27]. In contrast, SOC exhibited a high variation at both layers, with greater coefficients of variation (CV) of 43.81% and 37.59%, respectively. Likewise, similar patterns were observed in TN at the depth of 0–20 cm and C:P at the depth of 20–40 cm.

**Table 2.** ANOVA and LSD test of soil C, N, P concentrations, eco-stoichiometry, and other physico-chemical properties at different depths (*n* = 46). SE, standard error; CV, coefficient of variation. The different letters in the same row indicate a significant difference at the 95% level according to the LSD test.

| Indicators | 0–20 cm | | 20–40 cm | | *p*-Value |
|---|---|---|---|---|---|
| | Mean ± SE | CV | Mean ± SE | CV | |
| SOC (g·kg$^{-1}$) | 20.08 ± 1.31 [a] | 43.81% | 15.67 ± 0.88 [b] | 37.59% | 0.0063 |
| TN (g·kg$^{-1}$) | 1.51 ± 0.09 [a] | 38.90% | 1.15 ± 0.06 [b] | 33.61% | 0.0009 |
| TP (g·kg$^{-1}$) | 0.9 ± 0.03 [a] | 26.06% | 0.8 ± 0.03 [a] | 29.08% | 0.0524 |
| C:N | 13.89 ± 0.64 [a] | 30.79% | 13.61 ± 0.33 [a] | 16.49% | 0.6996 |
| C:P | 23.15 ± 1.08 [a] | 31.15% | 20.49 ± 1.19 [a] | 38.81% | 0.1001 |
| N:P | 1.76 ± 0.07 [a] | 27.43% | 1.5 ± 0.08 [b] | 33.73% | 0.0167 |
| pH | 5.79 ± 0.09 [a] | 10.36% | 6.53 ± 0.08 [b] | 8.61% | 0.0000 |
| Sand (%) | 24.66 ± 2.00 [a] | 54.41% | 23.75 ± 2.21 [a] | 62.30% | 0.7607 |
| Clay (%) | 5.31 ± 0.35 [a] | 43.76% | 5.37 ± 0.32 [a] | 39.66% | 0.9030 |
| BD (g·cm$^{-3}$) | 1.3 ± 0.02 [a] | 9.98% | 1.39 ± 0.02 [b] | 7.50% | 0.0005 |
| SWC (g·kg$^{-1}$) | 0.22 ± 0.01 [a] | 24.23% | 0.25 ± 0.01 [b] | 22.29% | 0.0164 |

### 3.1.2. Geostatistical Analysis of Soil C, N, and P Concentrations and Eco-Stoichiometry

The best-fitted semivariogram models and parameters of soil C:N:P contents and their ratios at different depths are shown in Table 3. The classification method of spatial dependence for soil properties (i.e., <0.25, strong spatial variation; 0.25–0.75, moderate spatial variation; and >0.75, weak spatial variation) was adopted [28]. The results indicated that SOC showed a strong spatial autocorrelation at different soil depths, while TN showed a moderate degree of spatial autocorrelation at the depth of 0–20 cm and a strong spatial autocorrelation in the 20–40 cm layer. TP varied widely and showed a medium-degree spatial autocorrelation at different soil depths. The C:N ratio in the 0–20 cm soil layer showed a larger variation than in the 20–40 cm layer, with a moderate degree of spatial autocorrelation. The soil C:P and N:P values presented a moderate degree of spatial autocorrelation at the depth of 20–40 cm. All these results indicated that it was practical to interpolate soil C:N:P contents and their ratios in the study area.

**Table 3.** Models and parameters fitted to the semivariogram of soil C, N, and P concentrations and the eco-stoichiometry ratios at different soil depths.

| Indicators | Soil Layers/(cm) | Skewness | Kurtosis | Transformation | Models | C$_0$/(C$_0$ + C) | Range/(km) |
|---|---|---|---|---|---|---|---|
| SOC | 0–20 | 0.18 | 2.06 | lg | Exponential | 0.13 | 56.30 |
| | 20–40 | 0.11 | 2.88 | - | Spherical | 0.04 | 61.83 |
| TN | 0–20 | 0.10 | 3.42 | lg | Gaussian | 0.35 | 53.71 |
| | 20–40 | −0.24 | 2.31 | - | Gaussian | 0.04 | 53.70 |
| TP | 0–20 | −0.30 | 2.58 | lg | Gaussian | 0.33 | 59.13 |
| | 20–40 | 0.25 | 2.01 | lg | Gaussian | 0.52 | 46.60 |
| C:N | 0–20 | 1.45 | 6.50 | lg | Exponential | 0.33 | 69.90 |
| | 20–40 | 0.08 | 2.69 | - | Spherical | 0.20 | 65.63 |
| C:P | 0–20 | −0.11 | 2.57 | lg | Exponential | 0.26 | 70.00 |
| | 20–40 | 0.36 | 2.88 | - | Gaussian | 0.52 | 56.33 |
| N:P | 0–20 | −0.08 | 2.85 | lg | Exponential | 0.17 | 74.03 |
| | 20–40 | 0.08 | 2.71 | - | Gaussian | 0.62 | 122.66 |

The spatial distribution maps of these nutrient variables were interpolated by ordinary Kriging at two soil depths (Figure 2). Although the concentrations of SOC, TN, and TP were higher in the 0–20 cm than those in the 20–40-cm layer, the highest soil C, N, and P levels were found in the northeastern part of the study area and showed similar decreasing trends from northeast to southwest in different soil layers. Regarding the soil C:N and C:P ratio levels at the depth of 0–20 cm, the high-value areas were mainly found in the northeastern area of the typical black soil region, whereas their spatial distribution maps were relatively smooth at the depth of 20–40 cm. High values of the soil N:P ratio in different soil layers were found for the northern end of the typical black soil area, while the values were relatively homogeneous throughout the southern part, irrespective of the soil depth.

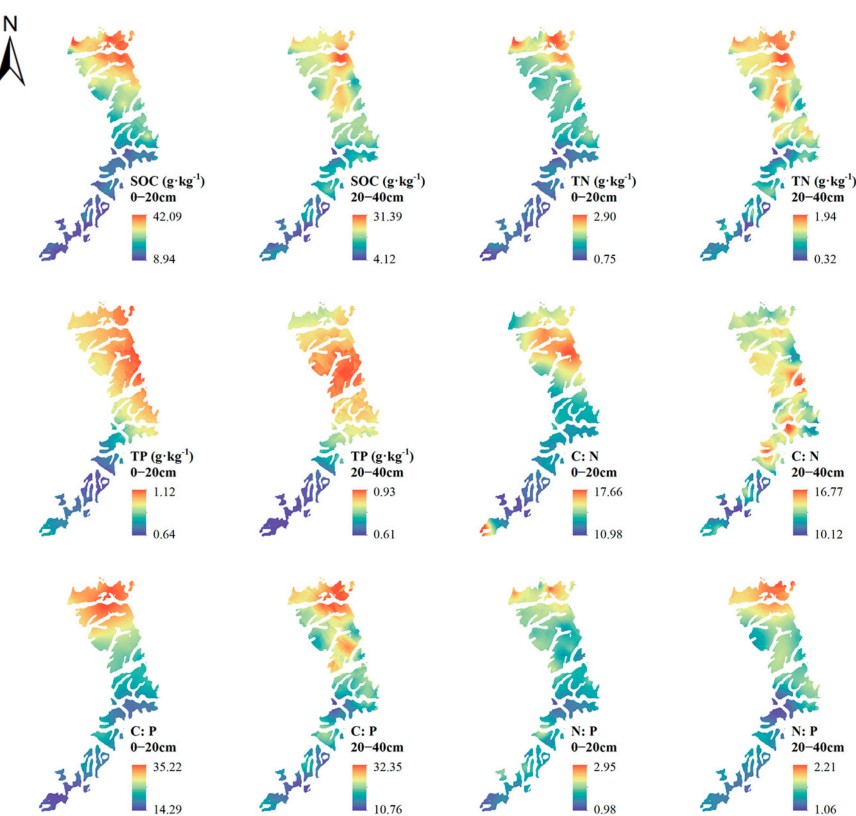

**Figure 2.** Spatial distribution of soil C, N, P concentrations and eco-stoichiometry in the typical black soil region of northeast China. SOC, soil organic carbon; TN, total nitrogen; TP, total phosphorus.

*3.2. Classification Cluster Analysis of Soil C, N, and P Eco-Stoichiometry Characteristics*

To properly discuss the characteristics of the eco-stoichiometry of soil C, N, and P at different soil depths, we used the square Euclidean distance to analyse the profile characteristics of the 46 samples based on the sum of square deviations. The clustering dendrogram was then established using hierarchical clustering (Figure 3). At a distance greater than 15, the sampling points could be divided into two clusters, where sampling sites exhibited a more apparent spatial clustering pattern than in three or four clusters. Moreover, the majority of indices to evaluate the compactness of clustering indicated that the optimal number of clusters was two (Table S1). The points in cluster 1, with higher soil C, N, and P levels as well as eco-stoichiometric ratios, were mainly located in the northeastern part of the study area. On the contrary, the points in cluster 2, with lower soil C, N, and P levels as well as eco-stoichiometric ratios, were distributed in the southwestern part of the study area (Figure 4). All the levels of soil C, N, C:N, C:P, and N:P in cluster 1 were higher than those in cluster 2 at different soil depths. The mean values of soil pH, BD,

and sand content in cluster 1 were lower than those in cluster 2. However, the mean value of SWC in cluster 1 was higher than that in cluster 2.

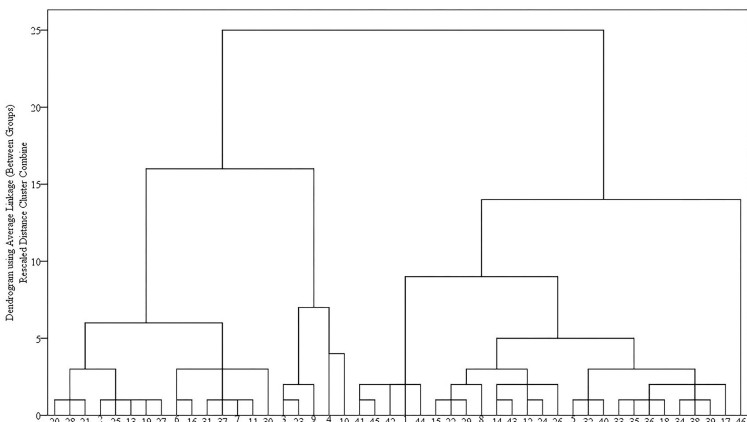

**Figure 3.** Hierarchical clustering of soil C, N, and P concentrations and eco-stoichiometry in the typical black soil region of northeast China. The numbers on the horizontal axis represent sample numbers.

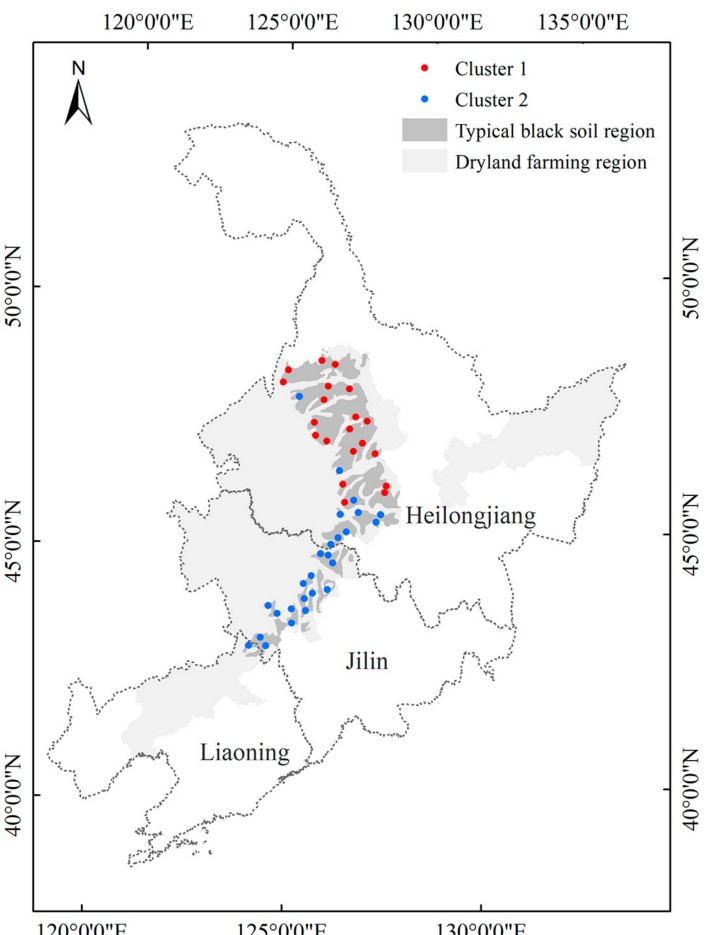

**Figure 4.** Hierarchical cluster plots of soil C, N, and P concentrations and eco-stoichiometry in the typical black soil region.

### 3.3. Effects of Environmental Variables on Soil C:N:P Eco-Stoichiometry

3.3.1. Correlation Analysis

Figure 5 shows the relationships between the soil C:N:P eco-stoichiometry and environmental variables in the study area. SOC and TN presented similar relationships

with latitude (Lat), BD, and SWC across different layers. SOC and TN showed positive correlation with Lat and SWC, and negative correlation with BD across all layers, except the 0–20 cm layer in cluster 2. Moreover, SOC and TN also positively correlated with Silt, but negatively correlated with Sand in the 0–20 cm layer of cluster 1 and the 20–40 cm layer of cluster 2. In addition, C:P and N:P ratios were also positively correlated with SWC and negatively correlated with BD in cluster 1, while in cluster 2, their negative correlation with BD was observed only in the 20–40 cm layer. In terms of TP, it was rarely correlated with the environmental variables in all layers, except the 20–40 cm layer in cluster 2, in which it was positively correlated with Lat and Hum, but negatively correlated with MAT, MAP, and MCI. C:N ratio was not closely correlated with any environmental variables in cluster 1, while in cluster 2, it was negatively correlated with BD at the depth of 0–20 cm, and negatively correlated with MAT and elevation at the depth of 20–40 cm.

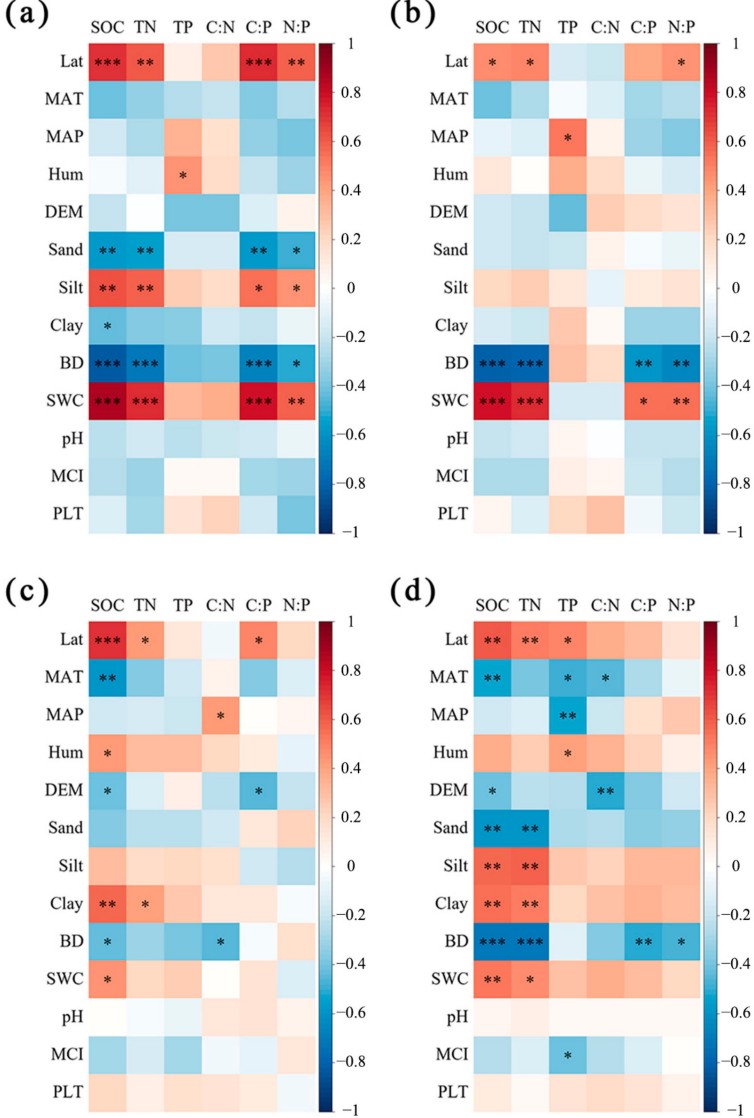

**Figure 5.** Heat maps for the correlation coefficients between soil C, N, and P contents and eco-stoichiometric ratios and environmental variables at the depths of (**a**) 0–20 cm, (**b**) 20–40 cm in cluster 1, and (**c**) 0–20 cm, (**d**) 20–40 cm in cluster 2. Stars indicate significance levels (* $p < 0.05$; ** $p < 0.01$; *** $p < 0.001$). SOC, soil organic carbon; TN, total nitrogen; TP, total phosphorus; Lat, latitude; MAT, mean annual temperature; MAP, mean annual precipitation; Hum, relative atmospheric humidity; DEM, digital elevation model; BD, bulk density; SWC, soil water content; MCI, multiple cropping index; PLT, plough layer thickness.

### 3.3.2. Contribution of Environmental Variables to Soil C:N:P Eco-Stoichiometry

The effects of environmental variables on soil C:N:P eco-stoichiometry in both clusters were analysed using RDA (Figure 6). The eigenvalues indicated that most of the variances at different depths (0–20 cm and 20–40 cm) were explained by axis 1 in clusters 1 and 2. The total variation explained by all variables on axis 1 and 2 at different soil depths was 69.4% and 71.8% in cluster 1, and 45.1% and 67.8% in cluster 2, respectively. At the depth of 0–20 cm, SWC had the highest effect on soil C:N:P eco-stoichiometry in cluster 1, which was 62.9% ($p < 0.01$), while Lat and MAP had a higher effect on soil C:N:P eco-stoichiometry in cluster 2, which was 17.2% ($p < 0.01$) and 9.4% ($p < 0.05$), respectively. At the depth of 20–40 cm, BD had the highest effect on soil C:N:P eco-stoichiometry in cluster 1, which was 37.4% ($p < 0.01$), while BD, DEM, and MAP had a higher effect on soil C:N:P eco-stoichiometry in cluster 2, which was 29.0% ($p < 0.05$), 12.4% ($p < 0.01$) and 10.6% ($p < 0.01$), respectively. Overall, MAP was one of the main factors in cluster 2. BD at the 20–40 cm depth was the most significant controlling factor in cluster 1 and 2.

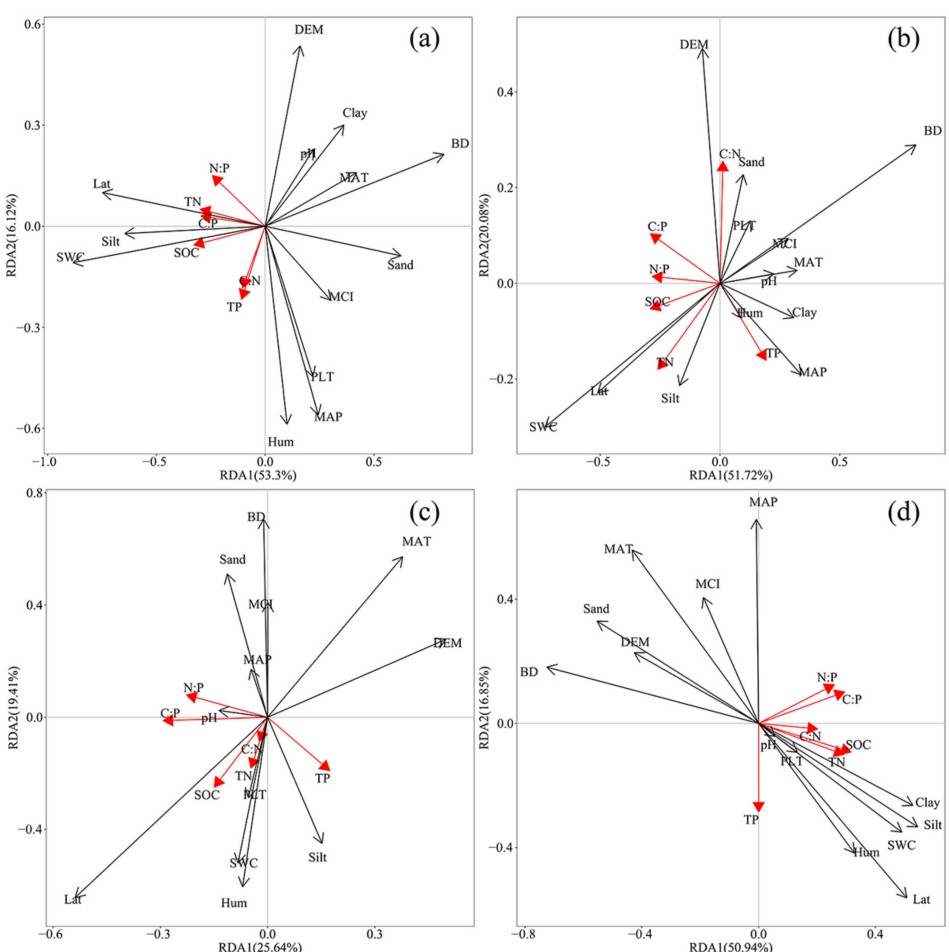

**Figure 6.** Redundancy analyses of soil C:N:P eco-stoichiometry and environmental variables at the depths of (**a**) 0–20 cm, (**b**) 20–40 cm in cluster 1, and (**c**) 0–20 cm, (**d**) 20–40 cm in cluster 2. SOC, soil organic carbon; TN, total nitrogen; TP, total phosphorus; Lat, latitude; MAT, mean annual temperature; MAP, mean annual precipitation; Hum, relative atmospheric humidity; DEM, digital elevation model; BD, bulk density; SWC, soil water content; MCI, multiple cropping index; PLT, plough layer thickness.

## 4. Discussion

### 4.1. Stoichiometric Characteristics of Soil C, N, and P

The soil C, N, and P eco-stoichiometric distributions are important, with respect to both agricultural sustainability and environmental protection. This study contributed to

the knowledge of their distribution related to natural conditions and agricultural activities, providing the basis for conservation management which could be extended to other black soil regions under similar conditions beyond the study area.

The decomposition of organic matter from plant residues is a main source of soil C. Biological fixation, precipitation, and fertilizer also contribute to soil N [29]. The capacity of these inputs decreases with increasing soil depth, which explains the patterns in which SOC and TN appeared to be significantly higher in the upper layer. The moderate spatial autocorrelation and the higher variation of soil N at 0–20 cm also indicated that it was more easily disturbed by external factors compared to the deeper layer. Additionally, parent materials supplement the underlying TP by weathering, resulting in a relatively stable vertical distribution of TP [5]. Compared to the overall levels of China, the mean contents of soil C, N, P in the study area were much higher, which is in line with the previous studies [2,30].

Soil C:N ratio influences the decomposition rate of organic matter by microorganisms, thus affecting the cycle of SOC and N [31]. The mean C:N ratio in both layers was slightly lower than that for China, mainly due to higher microbial activity in decomposition triggered by fertilizer application as well as the addition to soil nitrogen content [32]. In addition, the lack of straw returning may also be responsible for it, which is a source of organic matter. The C:P and N:P ratios were significantly smaller than those for the whole country; this can be explained by the high intrinsic P level of our study area [2]. In addition, other agriculture activities (e.g., land use changes, tillage) also significantly changed the vertical distribution and decomposition rates of C and N, resulting in fluctuations in soil C, N, P ratios within a specific depth range [33,34].

Consistent with previous studies, the contents and ratios of soil C, N, P at different depths decreased from northeast to southwest in the study area. It is probable that temperature and precipitation were the main factors affecting the spatial patterns of soil C:N:P eco-stoichiometry characteristics at a large regional scale, as reported by previous studies [18,35,36]. Under the condition of unchanged precipitation and other soil-forming factors, SOC content in temperate regions increases with decreasing temperatures, as a result of inhibited decomposition of soil organic matter [20,37]. Meanwhile, with a lower temperature in the northeast part, the lower N requirement from weaker microbial activities consequently promotes the accumulation of soil N. The spatial pattern of TP can be explained by the organic phosphorus variation, which was affected by SOC distribution [38].

Based on our results, the sampling points in the study area could be divided into two clusters, with the Songhua River as an important transition zone. One possible explanation is that the mean annual temperatures on the two sides of this transition zone are different, albeit without differences in precipitation, as reported by a previous study [39]. In addition, the cropping system north of the zone was dominated by maize-soybeans, while south of this zone, maize was the main crop, potentially indicating different root biomass inputs, tillage methods, and thus decomposition rate of organic matter [40]. On both sides of the transition zone, differences in temperatures and cultivation can result in a different soil C:N:P eco-stoichiometry; however, the underlying mechanism still needs to be investigated.

*4.2. The Differences in Influencing Factors*

In this study, the influencing factors of soil C, N, P eco-stoichiometry differed across the clusters. The SWC level was relatively high in cluster 1, and its contribution to the eco-stoichiometry was more prominent than those of other soil properties in the 0–20 cm layer. High soil moisture favours the accumulation of chemical elements (e.g., C and N) rather than decomposition, especially under anaerobic conditions. Additionally, soil moisture has been reported to facilitate plant production and, consequently, enhance nutrient contents [41,42]. However, at the same depth in cluster 2, Lat was observed to be the most important controlling factor. It can be explained that soil C, N, and P stoichiometry were affected by land use history. Most of the samples in the cluster 2 were distributed in the southern part of the study area, with a relatively longer cultivation history. The long

cultivation time may affect SOC through its impact on soil microbial activities by tillage and other agricultural management [43]. Additionally, the removal of crop biomass in harvest can reduce C and N inputs back to the soil.

Meanwhile, at the depth of 20–40 cm, BD played a vital role in the soil C, N, and P contents and stoichiometry in both clusters. Cluster 1 featured higher values of eco-stoichiometry with a higher BD and a lower SWC, whereas cluster 2 exhibited the opposite pattern. It can also be explained by the cultivation history difference between two clusters, and a high BD always characterises low soil water permeability. The significant impact of BD in the deeper layer was mainly because the soil in the study area was affected by tillage and natural compaction. The transport and accumulation of soil C, N, and P are affected by the interaction among higher BD and lower SWC [22], which was more pronounced in the 20–40-cm layer. In this sense, root growth and litter decomposition were mostly affected by BD, and the migration and accumulation of elements in the soil was limited by high BD levels, thus inhibiting organic carbon and nitrogen production [44,45].

However, MCI and PLT, as the only two indicators of agricultural activities in this research, did not showed a significant impact on soil C, N, and P contents and stoichiometry. It is probable that these two indices varied in a narrow range in the study area, which partly agrees with the previous research [46]. In addition, it is worth noting that temperature did not act as an important controlling factor, which was the opposite of many previous studies [47,48]. It was possible that the mean value of a single year was not representative enough to show the temperature pattern in a relatively long period, or the effect of temperature was counteracted by other factors such as precipitation, as previous research reported [49]. In this study, the environmental variables listed only explained a limited part of the variation in soil C, N, and P contents and stoichiometry. It might be attributed to the absence of data from agricultural activities due to the collection limitations. In further study, these factors need to be better explored.

## 5. Conclusions

We provided a complete picture of the spatial patterns of the C, N, and P levels and the C:N:P ratios in farmland soil of a typical black soil region in northeast China. The soil C, N, and P concentrations and their ratios decreased with increasing soil depth. The eco-stoichiometry of soil C, N, and P showed a significant heterogeneity in the different soil depths, and presented an increasing trend from southwest to northeast. Systematic cluster analysis confirmed that the Songhua River was an important transition zone for two clusters, and the soil C, N, and P contents and C:N:P ratios in cluster 1, north of the transition zone, were higher than those in cluster 2, south of the transition zone.

The study also showed that the main factors affecting the values of soil C, N, and P contents and ratios differed laterally and vertically in the typical black soil area of northeast China. At the depth of 0–20 cm, SWC had the strongest explanatory ability for the evaluated soil parameters in cluster 1, and latitude had the highest explanatory ability in cluster 2. At the depth of 20–40 cm, BD was the dominant factor affecting both clusters. Our findings contribute to improved knowledge of C, N, and P dynamics and balance and their ecological interactions in northeast China.

**Supplementary Materials:** The following supporting information can be downloaded at: https://www.mdpi.com/article/10.3390/su14052636/s1, Table S1: Optimal values of clusters.

**Author Contributions:** Conceptualization, Z.Z.; methodology, Q.C.; formal analysis, Q.C.; investigation, Z.Z. and Y.G.; resources, Z.Z.; writing—original draft preparation, Q.C. and Z.Z.; writing—review and editing, S.C.; visualization, Q.C.; supervision, Z.S.; funding acquisition, Z.S. All authors have read and agreed to the published version of the manuscript.

**Funding:** This research was funded by the National Natural Science Foundation of China (U1901601) and the Ten-thousand Talents Plan of Zhejiang Province (2019R52004).

**Institutional Review Board Statement:** Not applicable.

**Informed Consent Statement:** Not applicable.

**Data Availability Statement:** Not applicable.

**Acknowledgments:** The authors are deeply grateful to the anonymous reviewers and the editor for their helpful comments on the manuscript.

**Conflicts of Interest:** The authors declare no conflict of interest.

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
