# Peer review of "Role of Environment Variables in Spatial Distribution of Soil C, N, P Ecological Stoichiometry in the Typical Black Soil Region of Northeast China"

_sustainability, doi:10.3390/su14052636_

Round 1
Reviewer 1 Report
The article "Role of environment variables in spatial distribution of soil C, 2 N, P ecological stoichiometry in the typical black soil region of 3 northeast China" is written well and a comprehensive study has done.
Author Response
Point 1: The article "Role of environment variables in spatial distribution of soil C, 2 N, P ecological stoichiometry in the typical black soil region of 3 northeast China" is written well and a comprehensive study has done.
Response 1: Thank you very much for the positive feedback.
Reviewer 2 Report
General comments:
In general, the work is well organized and described. Objectives and conclusion are not strong but the study deal with lots of field data and different analytical approaches. Here below specific comments.
Specific comments:
Lines 136-137: I suggest to mention firstly the ANOVA and then the LSD test as this is also the application order of the tests.
Line 150 (and Table 2): You have mention the ANOVA and the LSD test, please show the results of these tests (ie: p-value, standard error, mean value, significant letters) instead of the results reported in table 2 that are less informative.
Line 292: This sentence has no meaning here.
Line 299: But in table 1 you have indicated maize as crop for each sampling point. please be more clear about this.
In general, the discussion section needs to be improved considering all the results you have presented in the study. Consider also how your work could be of interest outside of the study area.
Author Response
We would like to thank the reviewer for the comments. The modification explanations are as follows.
Point 1: Lines 136-137: I suggest to mention firstly the ANOVA and then the LSD test as this is also the application order of the tests.
Response 1: Thank you for the suggestion. The order has been corrected.
Point 2: Line 150 (and Table 2): You have mention the ANOVA and the LSD test, please show the results of these tests (ie: p-value, standard error, mean value, significant letters) instead of the results reported in table 2 that are less informative.
Response 2: Thank you for pointing this out. We agree with this comment. The informantion of table 2 has been replaced accordingly by the results of ANOVA and the LSD test.
Point 3: Line 292: This sentence has no meaning here.
Response 3: Thank you for pointing this out. The sentence has been modified accordingly.
Point 4: Line 299: But in table 1 you have indicated maize as crop for each sampling point. please be more clear about this.
Response 4: I am sorry about the misuse of the word. The ‘corn’ means ‘maize’, which has been corrected now.
Point 5: In general, the discussion section needs to be improved considering all the results you have presented in the study. Consider also how your work could be of interest outside of the study area.
Response 5: Thank you for the comment. We have modified some sentences in the disscussion section and added more disscussion according to the results presented now. Additionally, the meaning of the work in other black soil regions has been considered accordingly.
Reviewer 3 Report
Introduction well written
- Materials and Methods
Line 85
Soybean is written twice
Figure 1. The orientation sign (North) and the grid are missing on the map and I suggest authors to locate the study area on the map of China
3 Results
Fig. 2 and Fig 4 Nothing on the figure indicates the north part of the study area
3.3.1. Correlation analysis
Illustrate the correlation description with figure numbers
Example
Line 226 – 227
SOC and TN showed positive correlation with Lat and SWC, and negative correlation with BD across all the layers (Fig. 5a, b and d), except the 0-20 cm layer in cluster 2 (Fig. 5c)
Line 236 – 237
…, while in cluster 2, it was negatively correlated with BD at the depth of 236 0-20 cm, and negatively correlated with MAT and elevation at the depth of 20-40 cm.
According to the Fig. 5d; C: N ratio was not negatively but positively correlated with MAT and elevation (DEM) at the depth of 20-40 cm. Please correct it.
Conclusion
The conclusion summarizes key findings in relation to the study objectives
Author Response
We would like to thank the reviewer for the comments. The modification explanations are as follows.
Point 1:
2 Materials and Methods
Line 85: Soybean is written twice.
Response 1: We are very sorry. The error has been corrected.
Point 2:
2 Materials and Methods
Figure 1. The orientation sign (North) and the grid are missing on the map and I suggest authors to locate the study area on the map of China
Response 2: Thank you for pointing this out. The orientation sign and the grid lines have been added, and the location of the study area in China has been mapped accordingly.
Point 3:
3 Results
Fig. 2 and Fig 4 Nothing on the figure indicates the north part of the study area
Response 3: Thank you for point this out. The orientation sign has also been added to these figures.
Point 4:
3.3.1. Correlation analysis
Illustrate the correlation description with figure numbers
Example:
Line 226 – 227: SOC and TN showed positive correlation with Lat and SWC, and negative correlation with BD across all the layers (Fig. 5a, b and d), except the 0-20 cm layer in cluster 2 (Fig. 5c)
Line 236 – 237: …, while in cluster 2, it was negatively correlated with BD at the depth of 236 0-20 cm, and negatively correlated with MAT and elevation at the depth of 20-40 cm.
Response 4: Thank you for the suggestion. The figure numbers have been added to the description.
Point 5:
3.3.1. Correlation analysis
According to the Fig. 5d; C: N ratio was not negatively but positively correlated with MAT and elevation (DEM) at the depth of 20-40 cm. Please correct it.
Response 5: In Fig. 5d, blue colour represents for the negative correlation, indicating that C:N ratio was negatively correlated with MAT and DEM.

Round 2
Reviewer 2 Report
Dear authors,
thank you for your work and for the responses to my comments.
I have no more comments for you.
Best regards.